# Outcome Analysis of the Use of Cerament^®^ in Patients with Chronic Osteomyelitis and Corticomedullary Defects

**DOI:** 10.3390/diagnostics12051207

**Published:** 2022-05-11

**Authors:** Marcel Niemann, Frank Graef, Sufian S. Ahmad, Karl F. Braun, Ulrich Stöckle, Andrej Trampuz, Sebastian Meller

**Affiliations:** 1Center for Musculoskeletal Surgery, Charité-Universitätsmedizin Berlin, Corporate Member of Freie Universität Berlin, Humboldt-Universität zu Berlin, and Berlin Institute of Health, 13353 Berlin, Germany; frank.graef@charite.de (F.G.); sufian@ahmadortho.com (S.S.A.); karl.braun@charite.de (K.F.B.); ulrich.stoeckle@charite.de (U.S.); andrej.trampuz@charite.de (A.T.); sebastian.meller@charite.de (S.M.); 2Julius Wolff Institute for Biomechanics and Musculoskeletal Regeneration, Berlin Institute of Health at Charité-Universitätsmedizin Berlin, 13353 Berlin, Germany; 3Department of Orthopedic Surgery, Hannover Medical School, 30625 Hannover, Germany; 4Department of Trauma Surgery, University Hospital Rechts der Isar, Technical University of Munich, 80333 Munich, Germany

**Keywords:** osteomyelitis, Cerament^®^, bony defect, bone infection

## Abstract

Background: Chronic osteomyelitis (OM) is a progressive but mostly low-grade infection of the bones. The management of this disease is highly challenging for physicians. Despite systematic treatment approaches, recurrence rates are high. Further, functional and patient-reported outcome data are lacking, especially after osseous defects are filled with bioresorbable antibiotic carriers. Objective: To assess functional and patient-reported outcome measures (PROM) following the administration of Cerament^®^ G or V due to corticomedullary defects in chronic OM. Methods: We conducted a retrospective study from 2015 to 2020, including all patients who received Cerament^®^ for the aforementioned reason. Patients were diagnosed and treated in accordance with globally valid recommendations, and corticomedullary defects were filled with Cerament^®^ G or V, depending on the expected germ spectrum. Patients were systematically followed up, and outcome measures were collected during outpatient clinic visits. Results: Twenty patients with Cierny and Mader type III OM were included in this study and followed up for 20.2 ± 17.2 months (95%CI 12.1–28.3). Ten of these patients needed at least one revision (2.0 ± 1.3 revisions per patient (95%CI 1.1–2.9) during the study period due to OM persistence or local wound complications. There were no statistically significant differences in functional scores or PROMs between groups. Conclusion: The use of Cerament^®^ G and V in chronic OM patients with corticomedullary defects appears to have good functional outcomes and satisfactory PROMs. However, the observed rate of local wound complications and the OM persistence rate may be higher when compared to previously published data.

## 1. Introduction

Chronic osteomyelitis (OM) is a low-grade infection that progressively destructs the affected osseous structures [1]. This disease is categorized according to the presumed mechanism of infection, such as hematogenous or continuity. For treatment strategy planning, the Cierny and Mader classification is most commonly used to grade OM subtypes [2]. This classification system addresses the infection location and takes patients’ comorbidities into account (localization: (1) medullary, (2) superficial, (3) localized, or (4) diffuse; physiological status: (A) normal host with no compromise, (B) local compromise, systemic compromise or local and systemic compromise, or (C) treatment worse than disease), and helps clinicians to decide on a treatment strategy. However, the management of chronic OM is highly challenging. Despite long-term antibiotic treatment following extensive surgical debridement, the recurrence rate is up to 30% within the first 12 months after surgery [3]. Therefore, standardized antibiotic concepts and surgical treatment strategies have been elaborated, which comprise single-stage and two-stage treatment protocols [4]. For this purpose, several ceramic- and non-ceramic-based products have been launched, some of which are additionally impregnated with antibiotics [5,6]. However, the need for secondary removal of most of these products negatively affects their broad adaptation to standard treatment strategies. In recent years, a calcium sulfate hydroxyapatite biocomposite, Cerament^®^ G or V (Bonesupport, Lund, Sweden), has been broadly advertised due to its high bioresorbable quality [7,8] and the locally increased bone formation [4,5,9]. Reported surgical outcomes [10] and patient satisfaction rates [11] following Cerament^®^ application are highly promising. Besides the surgical outcome, however, the functional outcome may similarly influence patients’ quality of life. Unfortunately, functional outcomes for different chronic OM locations have, thus far, not been assessed by previous authors.

Therefore, we performed a retrospective functional and patient-reported outcome (PROM) analysis of all patients operated on due to a chronic OM with a corticomedullary osseous defect.

## 2. Methods

We conducted a retrospective, single-center analysis of all patients who underwent surgery for a chronic OM with a corticomedullary defect between January 2015 and December 2020 at the Campus Virchow clinic of the Center for Musculoskeletal Surgery of the Charité—Universitätsmedizin Berlin. Local ethics committee approval was obtained (application number EA2/132/15, approved on 31 October 2015) and, due to the retrospective nature of the study, the need for secondary study consent was waived.

Patients’ data were sought from the electronic medical data system SAP (SAP ERP 6.0 EHP4, SAP AG, Walldorf, Germany) used at the study center. We only included patients who were treated surgically for a chronic OM and who received Cerament^®^ by reason of a corticomedullary osseous defect. Both Cerament^®^ G and Cerament^®^ V applications were included in this study. Chronic OM was defined as having symptoms for a minimum of six months with clinical and radiological features accompanied by at least one of the following: the presence of a sinus, an abscess or intra-operative pus, supportive histology, or at least two microbiological cultures with detectable organisms.

Patient characteristics obtained from the electronic medical system were age, gender, body mass index (BMI), the American Society of Anaesthesiologists physical status classification system (ASA), previous medical history, and pre-existing medical conditions assessed through the Charlson comorbidity index (CCI) [12]. We used the Cierny and Mader classification to grade OM subtypes [2]. Bony defects were measured digitally in Visage^®^ (Visage^®^ for Microsoft Windows, Version 7.1, Visage Imaging, Inc., Pro Medicus Limited, Richmond, Australia) using preoperatively assessed computed tomography. Lastly, using electronic medical records, we examined procedural extents and durations, microbiology results, length of stay (LOS), and postoperative antibiotic regimes.

At the study center, the standard protocol for the treatment of chronic OM comprises an intensive surgical debridement, an interdisciplinary coordinated antibiotic approach, and regular clinical follow-up visits at our outpatient clinic. For surgical treatment, a single-stage protocol is used [10,13]. Based on the individual patient case, this comprises sinus tract excision, preferably with primary wound closure, removal of osteosynthetic implants when radiographically or visually connected to the infection, and excision of any necrotic tissue, including removal of any sequestrum and intramedullary reaming, if appropriate. Corticomedullary defects are filled with Cerament^®^ G or V based on the presumed microbiologic results. In general, Cerament^®^ G is the primarily chosen antibiotic carrier due to its broader spectrum, as Gentamicin is active against most Gram-negative bacteria and Gram-positive *Staphylococcus*. Contrary, Vancomycin does not cover Gram-negative bacteria. Accordingly, Cerament^®^ V would only be chosen if previous microbiologic results ruled out an infection due to Gram-negative bacteria. No additional material or antibiotics are added locally. Regarding the antibiotic approach, we worked with an interdisciplinary team of infectious disease specialists. Antibiotic concepts are primarily based on the expected microbiologic spectrum and then adapted based on the microbiology results. In chronic OM, antibiotics are intravenously administered for 10–14 days and then continued orally for a total antibiotic treatment duration of 12 weeks. The follow-up visits are regularly scheduled during the primary hospital stay, with the first follow-up visit taking place within two weeks after hospital discharge. Depending on the patient’s status and the clinical course, further visits are individually planned. For functional assessments during these follow-ups, various clinical scores are used, depending on the compromised body region. Regularly used scores are the Disabilities of the Arm, Shoulder, and Hand (DASH) score, including the sport and work module [14], Knee Society Score (KSS) [15], and Foot and Ankle Outcome Score (FAOS) [16]. Commonly, the DASH score is used for chronic OM of the upper extremities, KSS for chronic OM near the knee (distal diaphysis, metaphysis, and epiphysis of the femur and proximal epiphysis, metaphysis, and diaphysis of the tibia), and the FAOS for chronic OM near to the foot (distal diaphysis, metaphysis, and epiphysis of the tibia and the foot). Further, the short-form health survey 36-item score (SF-36) [17] and the numeric rating scale (NRS) are used as general health and satisfactory indicators in all chronic OM patients. The domains of the DASH, KSS, FAOS, and SF-36 range from 0 to 100, respectively, with higher scores indicating better function/lower impairment in KSS, FAOS, and SF-36. Only in the DASH score do higher scores indicate greater functional impairment.

The primary outcomes of this study were the frequency of revision surgeries and the functional status. Statistical analysis was performed using GraphPad Prism (GraphPad Prism 9 for macOS, Version 9.3.0 (345), GraphPad Holdings, LLC, San Diego, CA, USA). Radar charts were created in Excel (Microsoft Excel 2016, Redmond, WA, USA). For independent sample comparisons, the Mann–Whitney *U* test was used for discrete and continuous variables and the chi-squared test for categorical variables. Due to the explorative study character, we did not adjust for multiple testing. Unless stated otherwise, discrete and continuous variables are represented as mean ± SD (95%CI) and categorical variables as frequencies (%). All *p*-values are two-tailed, and *p*-values ≤ 0.05 were considered statistically significant.

## 3. Results

A total of 20 patients underwent surgery for a chronic OM with the use of Cerament^®^ by reason of a corticomedullary bony defect. Four patients (20.0%) had been receiving antibiotic therapy for 22.0 ± 30.4 weeks (95%CI −53.6–97.6) prior to that revision surgery due to local infections at the operated site: three patients (15.0%) due to a chronic OM and one patient (5.0%) due to a peri-implant infection in that region. Seventeen patients (85.0%) already had multiple revision surgeries due to chronic OM in their past medical history. These interventions reached back 149.7 ± 168.2 months (95%CI 70.9–228.4). Table 1 displays the demographic characteristics of the study cohort. Further, Appendix A provides an extensive overview of all patients included in this study.

The primary cause of OM development was osteosynthesis due to fractures. Initially, seven patients (35.0%) had a low-energy trauma (all falls from standing height) leading to five closed fractures, one second-degree open fracture, and one rupture of the anterior cruciate ligament. These injuries needed open reduction and internal fixation (ORIF) in six and closed reduction and internal fixation (CRIF) in one patient. Four patients (20.0%) had a high-energy trauma (three road accidents, one gunshot injury) leading to two open fractures (there was no reproducible documentation in the other two cases). Another two patients (10.0%) had ORIF due to unknowingly arisen fractures, and one patient (5.0%) had an unknown elective surgery. Four patients (20.0%) had suffered from an atraumatic, hematogenous OM (one of these patients due to sepsis resulting from acute lymphoblastic leukemia). Lastly, there was no documentation in two patient cases (10.0%), and the patients did not remember the primary type of injury either. Table 2 summarizes the OM locations and the classifications according to Cierny and Mader.

All patients had a corticomedullary defect, as observed in the assessed sectional images. These defects comprised a volume of 6.2 ± 5.6 cm^3^ (95%CI 3.5–8.8 cm^3^), and there were no significant differences (*p* = 0.7) between patients who underwent revision (7.3 ± 7.0 cm^3^, 95%CI 2.2–12.3 cm^3^) and those who did not (5.1 ± 4.0 cm^3^, 95%CI 2.3–7.9 cm^3^) within the follow-up period. Due to these defects, 15 patients (75.0%) received Cerament^®^ G and 5 patients (25.0%) received Cerament^®^ V. Further periprocedural characteristics are displayed in Table 3.

Prior to the aforementioned surgical debridement, representative deep-tissue samples were collected for microbiology in all patients. In these, 1.5 ± 1.8 (95%CI 0.6–2.3) types of different germs were detected per person. This figure tended to be higher in patients who needed further revision (1.8 ± 2.2, 95%CI 0.2–3.4) than in those who did not (1.1 ± 1.2, 95%CI 0.2–2.0), but the difference did not reach significance (*p* = 0.6). With regard to the samples, *Staphylococcus aureus* was most commonly detected (20/110 deep tissue samples). Patients received intravenous antibiotic treatment for 18.9 ± 14.3 days (95%CI 12.2–25.6), followed by 29.7 ± 38.4 days (95%CI 11.7–47.7) of oral antibiotic treatment. The different germ strains detected in the samples and the accordingly chosen antibiotic treatment concepts are visualized in Figure 1.

After Cerament^®^ administration, patients were regularly followed up for 20.2 ± 17.2 months (95%CI 12.1–28.3). During that time, 10 patients needed revision surgery. Two of these had persistent OM with the occurrence of a fistula. They underwent revision 91 and 182 days after Cerament^®^ administration, respectively. Six patients had local wound problems, including wound dehiscence and prolonged secretion, which were revised 4 to 39 days after Cerament^®^ administration. One patient had local pain due to Cerament^®^ leakage outside of the bone and needed revision 150 days after surgery, and one patient underwent revision due to local wound problems at another center.

Functional scores and PROMs were collected for a total of 10 patients (50.0%), which included two humerus, one femur, five tibia, and two calcaneum OM cases. Six of these patients had undergone further revision surgeries and four had not. We did not observe any significant differences in outcome measures between groups. The specific functional scores and the PROMs are displayed in Figure 2. Additional metrics are displayed in Appendix A.

## 4. Discussion

This study comprises a cohort of 20 patients treated with Cerament^®^ G or V due to a corticomedullary osseous defect in chronic OM. To the best of our knowledge, this is the first study to provide functional scores and PROMs for such a patient cohort with a follow-up duration of up to 59 months after Cerament^®^ administration.

In recent years, several authors have worked on standardized concepts and treatment guidelines to improve the management and outcome of chronic OM. It has been shown that multidisciplinary bone infection units facilitate an advance, thereby reducing healthcare utilization [18]. To date, the combination of adequate antibiotic treatment, surgical debridement, and muscle flap coverage tends to be the most successful treatment concept [19]. Despite these well-established treatment regimes, a recent systematic review observed high heterogeneity among different centers’ treatment protocols [20]. Further, some approaches aimed to shorten the settled systemic antibiotic administration duration in chronic OM [21]. This heterogeneity of treatment concepts may be attributed to the high relapse rate of chronic OM [3,19,22] and physicians’ wish to improve patients’ outcomes. To achieve this, various risk factors of patients affected by OM need to be critically evaluated. Thus far, a total disease duration exceeding three months, endogenic or iatrogenic immunosuppression, and previous osteosyntheses were proven to be associated with a higher risk for OM relapse [19]. Despite previously existing presumptions, an age over 60 years did not lead to a higher relapse risk [23].

Further, the management of osseous defects [8] is highly challenging in chronic OM patients. Several products have been marketed as effective for use in the dead space management [24]. An early published product was polymethylmethacrylate (PMMA), which is universally used in orthopedic surgery. However, PMMA is not resorbable and entails the need for secondary removal to prevent biofilm formation and prolongation of infection [25]. Therefore, various resorbable calcium–sulfate-based antibiotic carriers have been developed [26], as these products avoid the need for secondary removal. One of the most frequently used antibiotic carriers is Cerament^®^, which is loaded either with Gentamycin (Cerament^®^ G) or Vancomycin (Cerament^®^ V). The high local antibiotic release of Cerament^®^ was not observed to be accompanied by corresponding side effects, but has been shown to effectively reduce the local bacterial count [27]. Further, it decreased persistent infection rates and increased the bone growth rate in rats treated for OM [28]. Subsequently, Cerament^®^ has proved to be an effective therapeutic in the treatment of OM associated with diabetic foot syndrome [9] and in benign and borderline bone lesions [29]. However, wound infection rates were observed to be up to 20%.

The largest case series observing the adoption of Cerament^®^ in chronic OM was provided by McNally et al. [10]. The authors reported a consecutive cohort of 100 Type III and IV OM according to Cierny and Mader with an eradication rate of 96%. Only 4% had a recurrence of OM, which occurred within 145 to 563 days after debridement and Cerament^®^ administration. Despite these astonishing surgical results, the authors reported prolonged wound leakage in 6% of their cohort for up to 11 weeks after surgery. This may be a result of the Cerament^®^ resorption process [7], in which up to 25% of patients have radiographic evidence of material leakage out of the osseous tissue [30]. In our study, the need for revision following Cerament^®^ administration was markedly higher compared to the aforementioned results, even though patients’ characteristics were comparable. One explanation may be that McNally et al. [10] did not count local problems, such as prolonged wound leakage or dehiscences, as complications. This may not be adequate when objectively rating Cerament^®^ and all issues attributed to its usage. From our perspective, the outcome data presented in the present paper appears to be more realistic as real-world data for most centers.

Thus far, only one group has reported PROMs after Cerament^®^ application [31]. In this study, Cerament^®^ G and V was administered in two-stage septic total knee and total hip arthroplasty replacements. The authors observed an infection recurrence rate of 5% along with satisfactory functional scores, such as KSS ranging from 72 to 86. With correspondence to the SF-36, the 12-item Short Form Survey reached values of 69 to 100. Other authors observed that up to 53% of the patients were restored to a state of full or almost full health within the first year after surgery, considering surgical chronic OM management alone [11]. In our study, functional scores were highly heterogeneous regarding the different scales. Especially, scores assessing patients’ sport functions and expectations (DASH sport module, KSS satisfaction and expectation, and FAOS sport) were beyond 50% of the potentially achievable score range. Only the KSS functions exceeded the 75th percentile of the achievable score range. Global health scores tended to be higher, indicating a low health impairment. Most domains of the SF-36 exceeded the 50th percentile of the achievable score range. We observed only minor differences between patients who needed further revisions and those who did not. Almost all subscores of the SF-36 were higher in patients who did not undergo additional revision surgeries. Concerning the functional scores, DASH and FAOS revealed better scores in patients who did not undergo further revisions, while KSS was lower in these patients. None of these differences reached significance, although this may be attributed to the small sample size.

This study has both strengths and limitations. This is the first study to systematically provide functional scores and PROMs for chronic OM patients who received Cerament^®^ due to corticomedullary defects. These outcome measures are exceptionally rare but are of the utmost importance in the realistic measurement of patient outcomes and daily life functions. Further, chronic OM treatment followed a clear and consistent treatment concept without strategy changes within the study period. On the other hand, the study cohort is rather small, thereby limiting the power of statistical analysis. However, most previous studies examining the usage of calcium–sulfate-based antibiotic carriers in chronic OM were comparably small [32,33,34,35,36,37,38]. Future studies need to address the cohort size in order to sufficiently provide recommendations on the usage of bone void fillers. Further, risk factors for poor outcomes need to be identified in order to sufficiently improve patients’ outcomes. Nonetheless, outcome results of Cerament^®^ administration in chronic OM patients are highly relevant as they facilitate data comparability between different study centers. Otherwise, reported results of single centers are difficult to interpret regarding their global validity.

## 5. Conclusions

This is the first study to systematically assess outcome data of Cerament^®^ G or V usage in chronic OM patients with corticomedullary defects for up to 59 months. With overall good functional measures and PROMs, Cerament^®^ is a safe approach in such cohorts. However, patients need to be informed about the substantial risk of local wound complications and OM persistence after Cerament^®^ usage.

## Figures and Tables

**Figure 1 diagnostics-12-01207-f001:**
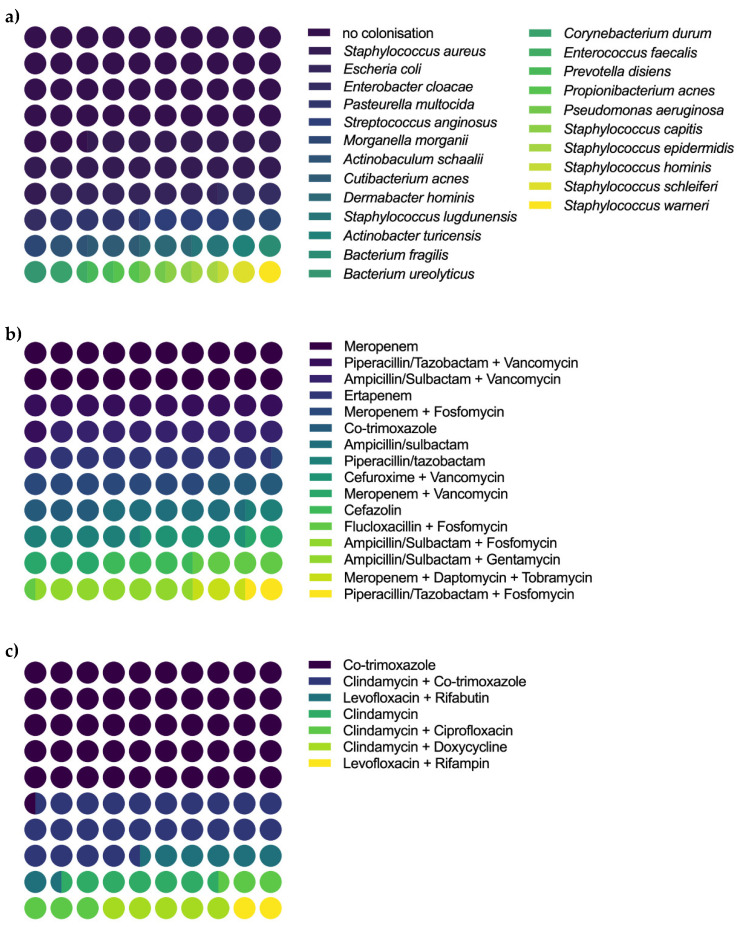
Colonization in microbiologic samples and intravenous and oral antibiotic regimes. (**a**) Depicts the microbiologic results of intraoperative assessment of deep tissue samples. There was no colonization in 47/110; *Staphylococcus aureus* in 20/110; *Escherichia coli* in 7/110, *Enterobacter cloacae*, *Pasteurella multocida* and *Streptococcus anginosus* in 4/110; *Morganella morganii* in 3/110, *Actinobaculum schaalii*, *Cutibacterium acnes*, *Dermabacter hominis* and *Staphylococcus lugdunensis* in 2/110 and *Actinobacter turicensis*, *Bacterium fragilis*, *Bacterium ureolyticus*, *Corynebacterium durum*, *Enterococcus faecalis*, *Prevotella disiens*, *Propionibacterium acnes*, *Pseudomonas aeruginosa*, *Staphylococcus capitis*, *Staphylococcus epidermidis*, *Staphylococcus hominis*, *Staphylococcus schleiferi,* or *Staphylococcus warneri* in 1/110 samples. (**b**) Patients received the following intravenous antibiotics: Meropenem for 74 days, Piperacillin/Tazobactam + Vancomycin for 39 days, Ampicillin/Sulbactam + Vancomycin for 37 days, Ertapenem for 32 days, Meropenem + Fosfomycin for 27 days, Co-trimoxazole for 22 days, Ampicillin/sulbactam or Piperacillin/tazobactam for 21 days, Cefuroxime + Vancomycin for 16 days, Meropenem + Vancomycin for 15 days, Cefazolin or Flucloxacillin + Fosfomycin for 14 days, Ampicillin/Sulbactam + Fosfomycin or Ampicillin/Sulbactam + Gentamycin for 1 day, Meropenem + Daptomycin + Tobramycin for 8 days or Piperacillin/Tazobactam + Fosfomycin for 5 days. (**c**) Patients received the following oral antibiotics: Co-trimoxazole for 299 days, Clindamycin + Co-trimoxazole for 142 days, Levofloxacin + Rifabutin for 42 days, Clindamycin for 36 days, Clindamycin + Ciprofloxacin for 33 days, Clindamycin + Doxycycline for 30 days or Levofloxacin + Rifampicin for 11 days.

**Figure 2 diagnostics-12-01207-f002:**
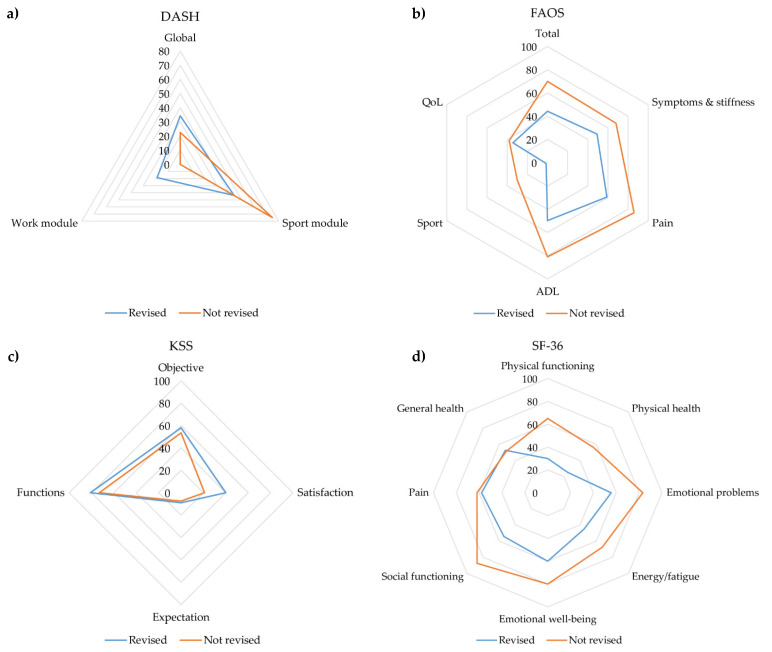
Functional outcome measures and PROMs of the study cohort. (**a**) Depicts the DASH, (**b**) the FAOS, (**c**) the KSS, and (**d**) the SF-36. Abbreviations: PROM: Patient-reported outcome measure, DASH: Disabilities of the Arm, Shoulder, and Hand Score, KSS: Knee Society Score, FAOS: Foot and Ankle Outcome Score, SF-36: Short-form Health Survey 36-item Score.

**Table 1 diagnostics-12-01207-t001:** Demographic overview of the study cohort.

	Total Sample (N = 20)	Revised (N = 10)	Not Revised (N = 10)	Statistic *
Gender (female [%]/male [%])	8 (40.0%)/12 (60.0%)	4 (40.0%)/6 (60.0%)	4 (40.0%)/6 (60.0%)	*p* > 0.9
age (years)	46.4 ± 16.3 (95%CI 38.8–54.0)	45.6 ± 17.0 (95%CI 33.4–57.8)	47.2 ± 16.5 (95%CI 35.4–59.0)	*p* = 0.7
BMI	24.7 ± 5.0 (95%CI 22.3–27.0)	25.1 ± 6.4 (95%CI 20.5–29.7)	24.2 ± 3.3 (95%CI 21.8–26.6)	*p* = 0.8
CCI	1.3 ± 1.8 (95%CI 0.5–2.1)	0.9 ± 1.0 (95%CI 0.2–1.6)	1.7 ± 2.3 (95%CI 0.1–3.3)	*p* = 0.7
ASA	1.8 ± 0.6 (95%CI 1.5–2.1)	1.8 ± 0.6 (95%CI 1.4–2.3)	1.8 ± 0.6 (95%CI 1.4–2.3)	*p* > 0.9
Previous revision surgeries	8.8 ± 15.7 (95%CI 0.4–17.1)	8.7 ± 12.3 (95%CI −4.2–21.5)	8.8 ± 18.1 (95%CI −4.1–21.7)	*p* = 0.8
Disease duration (months)	149.7 ± 168.2 (95%CI 70.9–228.4)	162.9 ± 180.6 (95%CI 33.7–292.1)	136.4 ± 163.5 (95%CI 19.4–253.4)	*p* = 0.9
Follow up (months)	20.2 ± 17.2 (95%CI 12.1–28.3)	21.0 ± 18.1 (95%CI 8.0–34.0)	19.4 ± 17.3 (95%CI 7.1–31.8)	*p* = 0.9

* Statistical analysis comparing revised and not revised patients. Abbreviations: BMI: Body mass index, CCI: Charlson comorbidity index, ASA: American Society of Anaesthesiologists’ physical status classification system.

**Table 2 diagnostics-12-01207-t002:** Location and type of chronic osteomyelitis.

Location	Total Sample	OM stage According to Cierny and Mader											
		Total Sample				Revised				Not Revised			
		III A	III B^L^	III B^S^	III B^LS^	III A	III B^L^	III B^S^	III B^LS^	III A	III B^L^	III B^S^	III B^LS^
Humerus	2	1	1	0	0	1	0	0	0	0	1	0	0
Ulna	1	0	0	0	1	0	0	0	0	0	0	0	1
Femur	3	1	0	1	1	0	0	1	0	1	0	0	1
Tibia	11	5	2	4	0	1	1	3	0	4	1	1	0
Calcaneum	3	1	0	2	0	1	0	2	0	0	0	0	0
Total	20	8	3	7	2	3	1	6	0	5	2	1	2

Abbreviations: OM: Osteomyelitis, III A: Localized osteomyelitis in a normal host, III B^L^: Localized osteomyelitis in a locally compromised host, III B^S^: Localized osteomyelitis in a systemically compromised host, III B^LS^: Localized osteomyelitis in a locally and systemically compromised host.

**Table 3 diagnostics-12-01207-t003:** Perioperative characteristics of the study cohort.

	Total Sample (N = 20)	Revised (N = 10)	Not Revised (N = 10)	Statistic *
Surgery duration (minutes)	114.7 ± 54.0 (95%CI 89.5–139.9)	107.6 ± 49.6 (95%CI 72.1–143.1)	121.8 ± 59.8 (95%CI 79.1–164.5)	*p* = 0.7
LOS (days)	21.0 ± 12.7 (95%CI 15.0–26.9)	21.4 ± 9.8 (95%CI 14.4–28.4)	20.5 ± 15.7 (95%CI 9.3–31.7)	*p* = 0.5
Intravenous antibiotics (days)	18.9 ± 14.3 (95%CI 12.2–25.6)	16.2 ± 9.3 (95%CI 9.6–22.9)	21.6 ± 18.1 (95%CI 8.7–34.5)	*p* = 0.6
Oral antibiotics (days)	29.7 ± 38.4 (95%CI 11.7–47.7)	24.6 ± 50.6 (95%CI −11.6–60.8)	34.8 ± 22.4 (95%CI 18.8–50.5)	*p* = 0.1
Following revisions	2.0 ± 1.3 (95%CI 1.1–2.9)	2.0 ± 1.3 (95%CI 1.1–2.9)	n. a.	n. a.
Duration until revision (days)	60.1 ± 65.7 (95%CI 9.6–110.6)	60.1 ± 65.7 (95%CI 9.6–110.6)	n. a.	n. a.

* Statistical analysis comparing revised and not revised patients. Abbreviations: LOS: Length of stay, n. a.: not applicable.

## Data Availability

The data presented in this study are available on request from the corresponding author. The data are not publicly available due to regulations of the local institutional ethics board.

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
