# Peer review of "Outcome Analysis of the Use of Cerament® in Patients with Chronic Osteomyelitis and Corticomedullary Defects"

_diagnostics, 2022, doi:10.3390/diagnostics12051207_

Round 1

Reviewer 1 Report

This article aims to present an Outcome Analysis of the Use of Cerament® in Patients with Chronic Osteomyelitis and Corticomedullary Defects.

The manuscript is written in good English and the clarity of expression, communication of ideas is also good. Although, the text is very generic and needs to be complemented by a more profound presentation and discussion of cases. References should be updated and modernized.

In the abstract and at other parts of the paper it was referred that twenty patients (eight of them female) with Cierny and Mader type III OM were included in this study and followed up for 20.2 ± 17.24 months after Cerament® administration. The authors should explain

  • the rationale for indicating the number of women
  • if the defects are identical and distributed in similar places in the body

making the comparison between cases statistically significative.

Authors should also discuss and account to the comparison why Cerament V or G is used.

In almost ALL the results and discussion presented, there is no care about the significative number of average and standard deviation, 20.2 ± 17.24 (line 25), 2 ± 1, (line 27), table 2, table 3, …

In line 55 it is claimed that a calcium sulphate hydroxyapatite biocomposite, Cerament® G or V (Bonesupport, Lund, Sweden), has been broadly advertised due to its local osteoconductive impact and its high bioresorbable quality, but Cerament is not a biocomposite and instead is a cement. What is intended to be understanded by local osteoconductive impact? In line 57 what is intended to be understanded by surgical outcomes highly promising and that they improve patient satisfaction through prolonged disease-free periods? So, with this material the disease is not eradicated? Still, patients and families and health systems are satisfied?

Table 4 presents a huge amount of information of clinical outcome measures of the study cohort, and my suggestion is to replace the table by graphs (pie, column, …), so the information is better cached by the reader.

In conclusion the article needs to be reformulated, complemented with more cases and as suggested before, there are some points that should be clarified before publication.

Author Response

Dear reviewer 1,

thank you for your review of our submitted manuscript and for the potential improvements you pointed out. Please find our detailed answers to your recommendations down below.

Reviewer: “This article aims to present an Outcome Analysis of the Use of Cerament® in Patients with Chronic Osteomyelitis and Corticomedullary Defects. The manuscript is written in good English and the clarity of expression, communication of ideas is also good. Although, the text is very generic and needs to be complemented by a more profound presentation and discussion of cases. References should be updated and modernized. In the abstract and at other parts of the paper it was referred that twenty patients (eight of them female) with Cierny and Mader type III OM were included in this study and followed up for 20.2 ± 17.24 months after Cerament® administration. The authors should explain the rationale for indicating the number of women if the defects are identical and distributed in similar places in the body making the comparison between cases statistically significative.”

Answer: Thank you very much for this comprehensive assessment of our manuscript. As suggested in this and in the following comments, we have worked on the manuscript and were able to substantially improve it. As you correctly observed, we reported data of 20 patients with type III osteomyelitis. We had included the total number of female patients to give future readers a hint on the gender distribution. But, this information does not have any impact on our actual study finding. Additionally, we had already included this information in Table 1. In order to make things clearer and to avoid double reporting, we have deleted the information in line 129 and solely kept the gender distribution in Table 1.

Reviewer: “Authors should also discuss and account to the comparison why Cerament V or G is used.”

Answer: This is a highly relevant remark, as Cerament® G and Cerament® V are different products. While Cerament® G is loaded with Gentamycin, Cerament® V is loaded with Vancomycin. In general, the antibiotic approach, including both the systemic and the local antibiotic therapy, should follow the suspected germ spectrum. If available, this may be based on previous microbiological findings. In our clinic, Cerament® G is the standard antibiotic carrier used to fill up corticomedullary defects in chronic osteomyelitis patients. Only if previous microbiological findings suggest potential resistances or a specific germ spectrum, Cerament® V is chosen. We updated the manuscript and included the specific process of decision-making (lines 95-99).

Reviewer: ”In almost ALL the results and discussion presented, there is no care about the significative number of average and standard deviation, 20.2 ± 17.24 (line 25), 2 ± 1, (line 27), table 2, table 3, …”

Answer: Thank you for your query. As correctly observed, we did not observe any significant difference between the patient revised and the patient not revised with regard to baseline demographics, perioperative surgical characteristics, and functional outcome data. We believe, that this is highly relevant to report, as such negative results shed light to a grey area of research. Accordingly, Tables 1, 3, and 4 display the inter-group comparison (revised vs. not revised) in the last column on the right, respectively. In the main text, we included respective p-values of these inter-group comparisons only if they were not already presented in a table. We believe that this is an appropriate way to report our results without double reporting informations.

Reviewer: “In line 55 it is claimed that a calcium sulphate hydroxyapatite biocomposite, Cerament® G or V (Bonesupport, Lund, Sweden), has been broadly advertised due to its local osteoconductive impact and its high bioresorbable quality, but Cerament is not a biocomposite and instead is a cement. What is intended to be understanded by local osteoconductive impact?“

Answer: Thank you for this remark, this is very relevant. Indeed, Cerament® are biodegradable antibiotic carriers based on calcium sulphate and 40 wt% Hydroxyapatite [1]. Depending on G or V, Cerament® is additionally loaded with either 175 mg per 100 ml Gentamycin or 66 mg per ml Vancomycin hydrochloride [1,2]. Contrary to bone cement, Cerament® is a biocomposite and resorption is generally observed by physicians [3]. This is one of the most relevant and highly advertised advantages of Cerament® compared to other local antibiotic carriers, as secondary surgeries for material removal are not necessary [4–6]. Accompanying the resorption, an increased bone formation has been observed [4], which is highly relevant in this cohort of corticomedullary defects patients. Additionally, we discussed different carriers in lines 237-250 in the manuscript. Nonetheless, we have rephrased the cited sentence to avoid misunderstanding.

Reviewer: “In line 57 what is intended to be understanded by surgical outcomes highly promising and that they improve patient satisfaction through prolonged disease-free periods? So, with this material the disease is not eradicated? Still, patients and families and health systems are satisfied?”

Answer: Thank you for this query. We intended to say, that 1) surgical outcome reported after Cerament® were generally good (few revision, long-term disease free etc.) and 2) patients were happy. As we all know, the eradication of a chronic osteomyelitis may be a highly ambitious goal in most patients, but Cerament® allows for a prolongation of periods between the surgical need for further revisions. As our phrasing may have been unclear, we have corrected lines

Reviewer: “Table 4 presents a huge amount of information of clinical outcome measures of the study cohort, and my suggestion is to replace the table by graphs (pie, column, …), so the information is better cached by the reader.”

Answer: Thank you very much for this hint. Table 4 is, indeed, rather large. The difficulty was that scores were documented in dependence on the region affected by the chronic osteomyelitis (see lines 105-118 in the revised manuscript). Accordingly, scores were individually assessed and, further, we tried to focus on the comparison between revised and not revised patients, as previous research observed lower satisfaction rates connected to more revisions. To highlight the key observations, we have marked the superordinate sub-scores from each of the reported scores. With this, it will be easier for readers to focus on the most important findings in Figure 4. Alternatively, a graphical presentation could be chosen. However, as we aimed to compare the aforementioned patient groups, most bar charts are not appropriate, as the DASH, the KSS, and the FAOS had only one patient in at least one of the groups. On the other hand, pie charts may not be suitable to present such functional scores, as these scores may not be summarized as parts of a whole. In the following, we have exemplary included a scatter dot plot of the SF-36 and its sub-scores. However, we would suggest to include such a figure solely as a supplementary file and not as an alternative to Table 4, as the loss of information would be too severe from our point of view.

Supplementary Figure 1. Distribution of absolute SF-36 sub-scores. Each column represents one of the SF-36 sub-scores as scatter dot plots. The columns’ height represents the mean of each sub-score and the error bars indicate the standard error of the mean. Abbreviations: SF-36: Short-form Health Survey 36-item Score.

Reviewer: “In conclusion the article needs to be reformulated, complemented with more cases and as suggested before, there are some points that should be clarified before publication.”

Answer: Thank you very much for this remarked. In accordance with your other remarks, we have updated our manuscript. Unfortunately, this study was conducted as a retrospective study analyzing all patients who were treated with Cerament® due to a chronic osteomyelitis between 2015 and 2020. Prior to 2015, we did not use Cerament®. Further, in 2021 operating room time was remarkably reduced due to corona virus. Therefore, we are, unfortunately, not able to report more patients that we have already reported in our manuscript.

Again, thank you for your thorough review. All of your constructive points significantly helped to improve our manuscript.

Best regards

The authors

References:

1       Ferguson J, Diefenbeck M, Mcnally M. Ceramic Biocomposites as Biodegradable Antibiotic Carriers in the Treatment of Bone Infections. J bone Jt Infect. 2017; 2: 38–51 Im Internet: https://pubmed.ncbi.nlm.nih.gov/28529863/

2       CERAMENT® V Product Fact Sheet. 2020; Im Internet: https://www.bonesupport.com/wp-content/uploads/2021/03/936.-PR-0936-02-en-EU-2020-10-CERAMENT-V-Product-Fact-Sheet.pdf

3       McNally MA, Ferguson J, Lau ACK, Diefenbeck M, Scarborough M, Ramsden AJ, Atkins BL. Single-stage treatment of chronic osteomyelitis with a new absorbable, gentamicin-loaded, calcium sulphate/hydroxyapatite biocomposite: a prospective series of 100 cases. Bone Joint J. 2016; 98-B: 1289–1296 Im Internet: https://pubmed.ncbi.nlm.nih.gov/27587534/

4       Oliver RA, Lovric V, Christou C, Walsh WR. Comparative osteoconductivity of bone void fillers with antibiotics in a critical size bone defect model. J Mater Sci Mater Med. 2020; 31 Im Internet: https://pubmed.ncbi.nlm.nih.gov/32840717/

5       Evans RP, Nelson CL. Gentamicin-impregnated polymethylmethacrylate beads compared with systemic antibiotic therapy in the treatment of chronic osteomyelitis. Clin Orthop Relat Res. 1993; 37–42

6       Kanakaris N, Gudipati S, Tosounidis T, Harwood P, Britten S, Giannoudis P V. The treatment of intramedullary osteomyelitis of the femur and tibia using the Reamer-Irrigator-Aspirator system and antibiotic cement rods. Bone Joint J. 2014; 96-B: 783–788 Im Internet: https://pubmed.ncbi.nlm.nih.gov/24891579/

Reviewer 2 Report

Dr. Niemann et al. researched Cerament® G and antibiotics for chronic osteomyelitis treatment. This research work was conducted over five years period. Various parameters have been evaluated, including DAS, KSS, FAOS, and SF-36, and the results are fascinating. Furthermore, it is interesting to know the first study on the utilization of Cerement G and V. Ceramic was found to be safe. Moreover, the utilization of antibiotics significantly reduces infection.

Figure 1. microbial species names should be in italics.

Author Response

Dear reviewer 2,

thank you for your review of our submitted manuscript and for the potential improvements you pointed out. Please find our detailed answers to your recommendations down below.

Reviewer: “Dr. Niemann et al. researched Cerament® G and antibiotics for chronic osteomyelitis treatment. This research work was conducted over five years period. Various parameters have been evaluated, including DAS, KSS, FAOS, and SF-36, and the results are fascinating. Furthermore, it is interesting to know the first study on the utilization of Cerement G and V. Ceramic was found to be safe. Moreover, the utilization of antibiotics significantly reduces infection. Figure 1. microbial species names should be in italics.”

Answer: Thank you very much for your remark. We have corrected Figure 1 and labeled all species in italics.

Again, thank you for your thorough review.

Best regards

The authors

Round 2

Reviewer 1 Report

An improvement in the text was suggested as it seemed too generic, but the authors made minor changes and did not complement it with a more in-depth presentation and discussion of the cases. References have not been updated and modernised.
The authors did not explain or discuss the distribution of defects in the body or the differences between the defects treated with Cerament to make the comparison between cases statistically significant.
The authors have not yet corrected the number of significant figures of the mean and standard deviation, for example 20.2 ± 17.24 (line 25), 2 ± 1, (line 27), table 2, table 3, … are incorrect. It should be 20.2 ± 17.2 and cannot be compared to 2 ± 1. This was completely disregarded throughout the paper.
As previously mentioned, Table 4 presents a huge amount of information from clinical outcome measures of the study cohort, and my suggestion is to replace the table with graphs (pie, column, …), so that the information is better stored by the reader. The information has not been improved.
In conclusion, the article still needs to be reformulated, complemented with more cases and as suggested above, there are some points that must be clarified before publication.

Author Response

Dear reviewer 1,

thank you for your review of our submitted manuscript and for the potential improvements you pointed out. Please find our detailed answers to your recommendations down below.

Reviewer: “An improvement in the text was suggested as it seemed too generic, but the authors made minor changes and did not complement it with a more in-depth presentation and discussion of the cases. References have not been updated and modernised. The authors did not explain or discuss the distribution of defects in the body or the differences between the defects treated with Cerament to make the comparison between cases statistically significant.”

Answer: Thank you for your recommendations. We have improved the main text and added a comprehensive overview of all patients included in this study. Accordingly, supplementary Table 1 displays an in-depth characterization of all patients included in the presented analyses. This included the individual defect sizes and defect locations of all patients. Further, we modernized the references. Unfortunately, we are not able to provide further patient cases due to the retrospective study design. We have added this to the limitations’ section and discussed the limitation of the small cohort. Nonetheless, we believe that it is highly relevant to provide such data, as corticomedullary defects in chronic osteomyelitis are still highly challenging. Therefore, data publication is most relevant to share observations and to compare treatment concepts and related patients’ outcomes.

Reviewer: “The authors have not yet corrected the number of significant figures of the mean and standard deviation, for example 20.2 ± 17.24 (line 25), 2 ± 1, (line 27), table 2, table 3, … are incorrect. It should be 20.2 ± 17.2 and cannot be compared to 2 ± 1. This was completely disregarded throughout the paper.”

Answer: Thank you for this remark. We have corrected the number of significant figures in the text and the tables in order to improve consistency of data presentation.

Reviewer: “As previously mentioned, Table 4 presents a huge amount of information from clinical outcome measures of the study cohort, and my suggestion is to replace the table with graphs (pie, column, …), so that the information is better stored by the reader. The information has not been improved. In conclusion, the article still needs to be reformulated, complemented with more cases and as suggested above, there are some points that must be clarified before publication.”

Answer: Again, this is a very essential suggestion. We have moved Table 4 to the supplementary files and included a new figure (Figure 2) in the main manuscript. In this, the means of all functional scores are presented as radar charts. This significantly raises comprehensibility. Between 2015 and 2020, our center retrospectively examined all osteomyelitis cases treated with Cerament®. According to the classification mentioned, the type III cases were considered. Unfortunately, due to the limited number of patients, even at our center, it is not possible for us to present more cases. Nevertheless, it is a comparably large cohort with a long follow-up compared to previous studies examining the usage of calcium-sulfate based antibiotic carriers [32–38].

Again, thank you for your thorough review. All of your constructive points significantly helped to improve our manuscript.

Best regards

The authors

References:

1            Humm G, Noor S, Bridgeman P, David M, Bose D. Adjuvant treatment of chronic osteomyelitis of the tibia following exogenous trauma using OSTEOSET Ò-T: a review of 21 patients in a regional trauma centre.

2            Andreacchio A, Alberghina F, Paonessa M, Cravino M, De Rosa V, Canavese F. Tobramycin-impregnated calcium sulfate pellets for the treatment of chronic osteomyelitis in children and adolescents. J Pediatr Orthop Part B. 2019; 28: 189–195 Im Internet: https://journals.lww.com/jpo-b/Fulltext/2019/05000/Tobramycin_impregnated_calcium_sulfate_pellets_for.2.aspx

3            McKee MD, Wild LM, Schemitsch EH, Waddell JP. The use of an antibiotic-impregnated, osteoconductive, bioabsorbable bone substitute in the treatment of infected long bone defects: early results of a prospective trial. J Orthop Trauma. 2002; 16: 622–627 Im Internet: https://pubmed.ncbi.nlm.nih.gov/12368641/

4            Gitelis S, Brebach GT. The treatment of chronic osteomyelitis with a biodegradable antibiotic-impregnated implant. J Orthop Surg. 2002; 10: 53–60 Im Internet: https://journals.sagepub.com/doi/10.1177/230949900201000110

5            Qin CH, Zhou CH, Ren Y, Cheng GY, Zhang HA, Fang J, Tao R. Extensive eggshell-like debridement technique plus antibiotic-loaded calcium sulphate for one-stage treatment of chronic calcaneal osteomyelitis. Foot Ankle Surg. 2020; 26: 644–649

6            Sun HJ, Xue L, Wu C Bin, Zhou Q. Use of Vancomycin-Impregnated Calcium Sulfate in the Treatment of Osteomyelitis of the Jaw. J Oral Maxillofac Surg. 2017; 75: 119–128

7            Zhao Z, Wang G, Zhang Y, Luo W, Liu S, Liu Y, Zhou Y, Zhang Y. The effect of calcium sulfate/calcium phosphate composite for the treatment of chronic osteomyelitis compared with calcium sulfate. Ann Palliat Med. 2020; 9: 1821833–1821833 Im Internet: https://apm.amegroups.com/article/view/39268/html

Round 3

Reviewer 1 Report

The authors improved the paper according to the reviewer suggestions. For that reason my opinion is that  the manuscript can be accepted in the present form.